# A Secure Data Aggregation Algorithm Based on a Trust Mechanism

**DOI:** 10.3390/s24134352

**Published:** 2024-07-04

**Authors:** Changtao Liu, Jun Ye

**Affiliations:** 1School of Cyberspace Security, Hainan University, Haikou 570228, China; liuchangtao@hainanu.edu.cn; 2Key Laboratory of Internet Information Retrieval of Hainan Province, Haikou 570228, China

**Keywords:** underwater wireless sensor networks, trust mechanism, dynamic slicing, data aggregation

## Abstract

Due to the uniqueness of the underwater environment, traditional data aggregation schemes face many challenges. Most existing data aggregation solutions do not fully consider node trustworthiness, which may result in the inclusion of falsified data sent by malicious nodes during the aggregation process, thereby affecting the accuracy of the aggregated results. Additionally, because of the dynamically changing nature of the underwater environment, current solutions often lack sufficient flexibility to handle situations such as node movement and network topology changes, significantly impacting the stability and reliability of data transmission. To address the aforementioned issues, this paper proposes a secure data aggregation algorithm based on a trust mechanism. By dynamically adjusting the number and size of node slices based on node trust values and transmission distances, the proposed algorithm effectively reduces network communication overhead and improves the accuracy of data aggregation. Due to the variability in the number of node slices, even if attackers intercept some slices, it is difficult for them to reconstruct the complete data, thereby ensuring data security.

## 1. Introduction

In recent years, underwater wireless sensor networks (UWSNs) have found increasingly widespread applications in marine environmental monitoring, underwater resource exploration, and military reconnaissance. However, due to the uniqueness of the underwater environment, such as dynamic water flow changes, node mobility, and limited communication, traditional data aggregation schemes face numerous challenges [1]. Most existing data aggregation solutions do not adequately consider node trustworthiness, potentially allowing falsified data from malicious nodes to contaminate the aggregation process, thus affecting the accuracy of the aggregated results. Moreover, given the underwater environment’s dynamic nature [2], current solutions often lack the flexibility to handle situations like node movement and network topology changes, making it difficult to adapt to the complexity and uncertainty of an underwater environment. This significantly impacts the stability and reliability of data transmission. Although traditional security protection techniques can defend against external network attacks, they provide inadequate protection against internal network threats [3]. Additionally, because underwater sensor networks have limited resources, it is crucial to control the computational complexity of security mechanisms while ensuring security [4]. Although slicing technology can effectively defend against various malicious attacks, it also increases communication pressure, which in turn brings greater energy consumption [5]. However, the trust mechanism can evaluate the behavior of nodes to assign trust values, thereby determining the number of slices, in order to reduce communication losses. Therefore, it is necessary to design an appropriate slicing algorithm based on actual needs and the network environment for secure data aggregation.

This work suggests a secure data aggregation (TMSDA) algorithm that is based on trust mechanisms and addresses the previously listed problems. Through the implementation of an adaptive model for evaluating trust, this method continuously evaluates the actions of neighboring nodes, guaranteeing that only highly trustworthy nodes are allowed to take part in the data aggregate process. The program uses a slicing–mixing–aggregation strategy in the meantime, dynamically slicing transmission data according to nodes’ trust values. Nodes with high trustworthiness are assigned fewer slices to reduce their communication and computation overhead, while nodes with low trustworthiness are assigned more slices to strengthen the verification and monitoring of their data. Additionally, the size of data slices is determined based on communication distance during transmission, with smaller slices transmitted over longer distances, minimizing node energy consumption while ensuring reliable data transmission. This method decreases the quantity of communication data in the network, lowering energy consumption and thereby lengthening the network’s lifespan, in addition to ensuring the security and dependability of data transfer. The following are this paper’s primary contributions:(1)A cluster head selection strategy based on node density is proposed. By first determining the local density of each node in the network and then designating nodes with a high density as cluster heads, this technique minimizes communication distances between nodes, uses less energy, and enhances the network’s overall performance.(2)A dynamic slicing scheme based on trust value is proposed, where the number of slices for a node varies dynamically with its trust value. For nodes with higher trust values, it is generally believed that they can more reliably complete transmission tasks, so their slice count can be reduced. Conversely, for nodes with lower trust values, their slice count is increased. This strategy can ensure security while reducing network traffic and improving data transmission efficiency.(3)A strategy based on data transmission distance to determine the slice size is proposed. The main purpose is to enable cluster member nodes to divide the data they want to send into different-sized data slices to adapt to neighbor nodes at different distances. This ensures that the energy consumption of each slice is optimal during data transmission.

The remainder of this paper is structured as follows: The current data aggregation strategies based on slicing technology are introduced in Section 2. The network model and key distribution technique of the scheme that is suggested in this study are further explained in Section 3. We give a thorough explanation of the TMSDA algorithm’s specifics in Section 4. Next, we carry out simulation studies to confirm the TMSDA algorithm’s functionality in Section 5. In Section 6, we provide a summary of the findings and suggest avenues for future investigation.

## 2. Related Work

The slicing algorithm is a data aggregation scheme that is relatively simple to deploy, has low computational complexity, and high security. Choosing a suitable aggregating scheme and slicing technique is the main focus of this algorithm research. Theoretically, network security will increase with the number of node slices. Nevertheless, a lot of data slices may cause a sudden spike in network traffic and energy usage, which could shorten the network’s lifespan. It may also raise the likelihood of errors, collisions, and packet loss when sending data. As a result, depending on the real requirements and the network environment, an appropriate slicing algorithm must be designed for secure data aggregation.

For different slicing methods, He et al. proposed a data aggregation scheme called SMART, in which nodes use a fixed slicing method to slice and encrypt data, and then randomly select a group of nodes to send the slices to them. Nodes that receive data slices will mix these slices and then send them to the aggregation node. This method can effectively protect data privacy, but correspondingly, the communication overhead will be relatively large [6]. Li et al.’s EEHA scheme is an improvement over SMART, achieving efficient and precise data aggregation while ensuring data security and privacy, reducing communication costs, and extending network lifespan [7]. Liu et al.’s HEEPP scheme is an energy-efficient data aggregation method that reduces communication overhead through random slicing and restricting leaf node participation, though it lacks security against node collusion attacks, similar to issues with the EEHA scheme [8]. Li et al.’s ESMART scheme enhances energy efficiency by optimizing the distribution of slice quantity, prolonging the lifetime of wireless sensor networks while maintaining high efficiency in data aggregation [9]. Wang et al.’s D-SMART algorithm dynamically slices data based on data importance, reducing communication costs and energy consumption, enhancing aggregation accuracy and privacy protection by determining data importance through deviations and means of perception variables [10]. Memon proposed an efficient, secure, and privacy-protecting data aggregation method named ESPPA to address communication overhead issues in wireless sensor networks. This method employs an advanced “slice and mix” strategy to effectively reduce communication overhead while ensuring data security and privacy, thus extending the network’s lifecycle and outperforming existing SMART schemes in privacy protection and communication efficiency [11]. Han et al. proposed a data aggregation scheme called PSPDA. This scheme improves upon the limitations of the SMART algorithm by introducing probabilistic partitioning, data coefficients, and positive and negative factors, while also considering random switching times [12]. Sun et al. proposed an improved slicing and mixing aggregation algorithm, ESMART, which dynamically adjusts data slice quantity to reduce underwater wireless sensor network communication overhead while maintaining data security. Compared to the SMART algorithm, ESMART performs better in network performance and energy consumption [13]. To address the shortcomings of the SMART algorithm, Li et al. proposed the FTSMART algorithm. FTSMART optimizes data slicing and aggregation tree generation based on a fat-tree structure, enabling fixed time interval allocation for nodes and reducing transmission conflicts [14]. Hajian et al. presented CHESDA, a data aggregation technique that uses slice mixing technology to protect privacy, fuzzy logic to choose the best slices, and GNY logic to confirm the key authentication mechanism. When it comes to energy efficiency, security, and communication overhead, CHESDA outperforms SMART [15].

For different aggregation methods, Fang et al. proposed a cluster-based data aggregation scheme called CSDA. This scheme utilizes slice assembly technology, demonstrating a high degree of flexibility and practicality. It can dynamically adjust the number of data slices based on the network scale, thereby effectively reducing energy consumption [16]. Liu et al. proposed the EPPA protocol, which is an energy-saving data aggregation method that reduces the number of slices and communication overhead through a novel slicing mode. They also developed the MPPA protocol, which supports multiple aggregation functions. Both protocols perform well in terms of privacy and efficiency [17]. Zhang et al.’s BPDA model uses a special balanced slicing strategy and mixing technology to improve privacy protection, designing three schemes considering node degree and energy to balance privacy protection [18]. In addition to emphasizing data transmission security, Hua et al. introduced an adaptive slicing data aggregation system that creatively considers nodes’ remaining energy and distance from the base station to accurately determine the slice size for each node. This method enhances data slicing efficiency, reduces node energy consumption, prolongs network lifespan, maintains a good level of privacy protection, and improves wireless sensor network performance under limited node resources [19]. Zhang et al.’s TDSM mechanism is a trust-based dynamic slicing method to enhance wireless sensor network performance, selecting cluster heads through trust evaluation, albeit with the limitation that once selected, they cannot be updated even if trust values decrease [20]. Dou et al. proposed a data aggregation algorithm for wireless sensor networks. By dynamically selecting cluster head nodes, slicing data, and introducing false information interference, this algorithm effectively reduces data traffic and significantly improves the privacy of node data [21]. Sheena et al. designed an energy-efficient data aggregation network slicing technique, EENS-DA, based on deep learning, utilizing Conv-LSTM networks for slicing and tree-like data aggregation to improve slice effectiveness and accuracy while protecting network privacy [22]. Zhao et al. proposed an intelligent pricing-based intelligent PDA scheme (PDA-SP), incorporating intelligent pricing and packaging methods to achieve multi-fare, multi-functional statistics, and efficiency improvement [23]. Zhou et al. introduced the EPDA data aggregation algorithm, which offers both energy efficiency and privacy preservation. This innovative approach organizes the sensor network into a tree-like structure, with chains formed by connecting leaf nodes. A unique feature of EPDA is that only the data from the tail nodes of these chains undergoes slicing, ensuring privacy safeguards while also minimizing energy consumption [24]. Hannah et al. proposed a data aggregation method named CEDAP. This method employs a dual-layer design: the inner layer gathers cluster data directly, while the outer layer utilizes a tree structure to transmit cluster head data. This design serves to optimize data transmission and prolong the network’s lifespan [25]. Hegde and Kulkarni proposed an energy and security-aware data survivability solution that is highly suitable for unattended wireless sensor networks in harsh environments. This solution considers multiple factors such as energy consumption, delay, distance, communication overhead, inter-cluster distance, and intra-cluster distance to ensure secure data transmission [26]. Bharany et al. introduced the SS-GSO method, which constructs a fitness function incorporating various information sources to determine the optimal number of clusters, build clusters, and select suitable cluster heads. This method aims to balance network energy consumption, shorten transmission distance, and extend network lifetime [27]. Tian et al. proposed a centralized control clustering scheme for UASNs, known as CCCS. This scheme employs adaptive clustering based on node density, with an intra-cluster controller designated for each cluster. By optimizing the selection process of relay nodes and relay clusters, this approach seeks to achieve balanced energy usage and improve routing efficiency [28].

## 3. Network Model

In UWSNs, common network topologies include tree, cluster, and tree–cluster hybrid. The tree topology, with a root node as its core, connects multiple branches and leaf nodes, suitable for branch management. Its advantages include simplicity, easy expansion, and fault isolation, but a failure in the root node can affect the entire system. The cluster topology groups sensor nodes according to clustering algorithms, with each group forming a cluster consisting of a cluster head and member nodes. Cluster members send data to the cluster head, which then interacts with the base station. This approach can balance energy consumption. The tree–cluster hybrid topology combines the characteristics of the first two. The cluster heads do not directly interact with the base station but communicate through a tree structure rooted at the base station. Data is first aggregated by the cluster heads and then sent up the tree to the base station, improving energy efficiency.

In order to better manage nodes in the network, this research introduces a network topology structure that mixes tree and cluster topologies, considering the peculiarities of an underwater environment. Base stations, cluster member nodes, and cluster head nodes are the three main components of this structure. The base station is capable of powerful processing and has a wealth of electrical resources. It sits at the top of the topological structure and is responsible for collecting and processing data. With the exception of the base station, all other nodes in the network are arranged into different clusters, each of which is made up of a cluster head node and a number of cluster member nodes. The fundamental function of cluster member nodes is to gather and transmit data to the cluster head node. The cluster head node is responsible for both receiving data transmitted by cluster member nodes and executing data aggregation procedures. Once the data aggregation method is completed, the cluster head node must upload the aggregated data results to its upstream node, which is located downstream of the base station along a tree structure. This carefully designed network topology ensures efficient and secure data transmission in an underwater environment, as shown in Figure 1.

Before constructing network topology and secure transmission channels between nodes, pre-distribution of keys is crucial. This approach of pre-allocating keys to nodes can effectively reduce unnecessary communication overhead during secure connection establishment between adjacent nodes, thereby enhancing the overall security and communication efficiency of the network. This work proposes a new approach based on the random key distribution mechanism proposed by Eschenauer and Gligor [29]. The key distribution method consists of three basic steps: key pre-distribution, shared key discovery, and path key establishment. The system first determines the precise size and security needs of the network during the key pre-distribution phase, and then it builds a sizable key pool using this data. The system then determines how many key rings each sensor node should have stored in advance of deployment. The goal of these key rings is to become ready for future secure communication by using keys that were randomly chosen from the key pool. To start the shared key discovery phase, deployed sensor nodes broadcast a list of key IDs from their shared key.

The current analysis assumes that each sensor node has a key ring with a size of k. These keys are randomly selected from a key pool of size K. Using this configuration, we can calculate the odds that any two surrounding nodes share at least one key.
(1)$$P_{connect}=1-\frac{{\left((K-k)!\right)}^{2}}{(K-2k)!K!}$$

Then, the likelihood that other nodes possess the same key as the node pair and successfully eavesdrop on encrypted messages is calculated as follows:(2)$$P_{overhear}=\frac{k}{K}$$

From Formula (2), it can be seen that to make the communication link more secure and reliable, the number of keys, K, in the key pool should be larger, so as to ensure that the probability of eavesdropping on the communication link between nodes, $P_{overhear}$, is smaller, and the probability of privacy data leakage is smaller. On the contrary, if the number of keys, K, is very small, the success rate of attackers eavesdropping on the communication link will be higher, and the probability of privacy data leakage will be greater. Assuming that there are *K* = 10,000 keys in the key pool, and each node selects 200 keys. If the probability of any pair of nodes having the same key is $P_{connect}=98.3\%$, then the probability of not having the same key is 1.5%. If a node pair does not have the same key, the above-mentioned path key establishment method can be used to form a shared key between nodes. If these two nodes select a shared key, other nodes may also have this key, but this possibility is a very small number, namely $P_{overhear}=0.2\%$.

This approach for distributing random keys is ideal for large-scale sensor networks. It can effectively prevent eavesdropping attempts, maintain secure connection between nodes, and assure the security of data transmission by utilizing a limited number of keys.

## 4. TMSDA Algorithm

This work proposes a data aggregation algorithm that is based on a trust mechanism. It may be broadly classified into four stages. Figure 2 shows the specific algorithm flow.

(1)Dynamic clustering stage: At this stage, the algorithm determines each node’s local density throughout the network and uses that information to dynamically choose cluster head nodes. Once every cluster head node in the network has been selected, the cluster is formed by each cluster head node selecting cluster member nodes based on their trust value toward neighboring nodes.(2)Dynamic slicing stage: In this stage, cluster member nodes determine the number of slices based on the trust value assigned by the cluster head node. Nodes with higher trust values receive fewer slices. Simultaneously, the size of the slices is adjusted according to the transmission distance between nodes, with distant nodes receiving smaller slices and closer nodes receiving larger ones. Finally, each cluster member retains one data slice and encrypts the remaining slices before randomly sending them to other neighboring nodes within the cluster.(3)Data mixing stage: During this stage, A node builds a new mixed data packet by receiving encrypted data slices from other nodes, decrypting them with a shared key, and combining them with its own retained slice.(4)Data aggregation stage: In this stage, cluster member nodes encrypt mixed data packets with a shared key before sending them to the cluster head node. The cluster head node collects all encrypted data slices from cluster members, performs decryption and aggregation operations to generate the final aggregated output, and then uploads it layer-by-layer to the base station.

### 4.1. Dynamic Clustering

In underwater environments, the distribution of sensor nodes often exhibits a random and dynamic nature. These nodes may constantly change their positions due to the influence of water currents, the activities of marine organisms, or interference from other external factors. This dynamic change in position undoubtedly poses numerous challenges to the construction and maintenance of underwater wireless sensor networks. One significant issue is the notable difference in node density across various regions within the network. This uneven node distribution directly affects the efficiency and energy consumption of data transmission. To address this problem, we propose a cluster head selection strategy based on node density. The core idea of this strategy is to optimize the network topology by selecting nodes located in high-density areas as cluster heads, thereby improving data transmission efficiency and reducing energy consumption. Our approach comprehensively considers the local density information of nodes to ensure that the selected cluster heads can effectively shorten the data transmission distance between cluster members.

Prior to putting this technique into practice, we determine each node’s node density. The number of nearby nodes that are within a sensor node’s communication range is usually used to compute node density, a crucial indicator that assesses the concentration of other nodes around the node. If a node is surrounded by more neighboring nodes, its node density is relatively high. High node density implies closer distances between nodes, which offers significant advantages during data transmission. Shorter distance communication requires less transmission power, effectively reducing energy consumption while enhancing communication reliability.

Our technique first determines the density of each node and then chooses nodes in high-density areas to be the cluster heads. This strategy has several advantages. First off, because data may reach the cluster head and be transmitted to its destination more quickly, it helps minimize the number of hops needed for data transmission. Second, there can be a significant reduction in the energy usage during communication because of the shorter distance between the cluster head and cluster members. Lastly, our approach can also improve the network’s overall performance by improving the network architecture, which will make the network more stable and robust.

Once the cluster head is selected, it invites trustworthy nodes to join its cluster based on its trust evaluation of neighboring nodes. This dynamic cluster formation mechanism not only considers the density factor of nodes but also ensures the trustworthiness and security of cluster members. Through this approach, our method can better adapt to various complex situations in underwater environments, guaranteeing the safety and reliability of data transmission.

In summary, our proposed cluster head selection strategy based on node density is an effective solution that addresses the issues of uneven node distribution and high energy consumption during data transmission in underwater environments. The particular procedure is as follows:(1)Cluster head selection:

The density, $\rho_{i}$, of each node i in the network is expressed by finding the number of surrounding nodes that are within its communication range. We choose the node with the highest density as the initial cluster head node, $C_{1}$, based on the density, $\rho_{i}$, of each node in the network. To prevent the chosen cluster head from having an influence, the density of the other nodes must be updated after the first cluster head, node $C_{1}$, is chosen. The purpose of updating is to reduce the density of nodes adjacent to the selected cluster head, thereby reducing their likelihood of becoming the next cluster head and avoiding excessive concentration of selected cluster heads in the network.

(2)Density update:

For each node, other than the first cluster head node, $C_{1}$, we update the density based on its distance from the selected cluster head node, $C_{1}$. The updated density, $\rho_{i}{\;\;}^{\prime }$, can be expressed as follows:(3)$$\rho_{i}{\;\;}^{\prime }=\rho_{i}-\alpha \cdot f(d(i,C_{1}))$$
where α is an adjustment parameter used to control the extent of density reduction, and $f(d(i,C_{1}))$ is a distance-related attenuation function that decreases with increasing distance. In this paper, a step function of the following form is used to represent it:(4)$$f(d(i,C_{1}))=\left\{\begin{array}{ll} 1, & d(i,C_{1})\leq R\\ 0, & d(i,C_{1})>R \end{array}\right.$$
where R is the communication radius of node i. This means that if a node i is within the communication range of $C_{1}$, its density will decrease by a fixed amount, α; otherwise, the density remains unchanged.

(3)Iterative cluster head selection:

We reorder the remaining nodes based on the revised densities, $\rho_{i}{\;\;}^{\prime }$, then designate the node with the highest density as the subsequent cluster head, $C_{2}$. Until the required cluster head count is reached or there are no more available eligible nodes, we repeat the density update and cluster head selection procedure.

Through the above steps, the selection of all cluster heads in the network is completed. Afterwards, cluster heads recruit neighboring nodes within their communication range to complete the construction of the cluster. When selecting cluster members, cluster heads use the trust evaluation model to calculate the trust values of neighboring nodes. Neighboring nodes with low trust values are not invited to join the cluster, thereby reducing the impact of abnormal nodes on network operation. The cluster building process is shown in Figure 3.

In addition, as the network continues to operate, some cluster head nodes may consume too much energy, leading to their inability to continue serving as cluster heads and withdrawing from the network. Moreover, in an underwater environment, nodes are constantly moving due to the influence of water flow, resulting in dynamic changes in network topology, and new nodes need to participate in the network. Therefore, after a certain period of network operation, it is necessary to re-select cluster heads and establish clusters according to the above steps.

### 4.2. Dynamic Slicing

During the dynamic slicing process, cluster head nodes play a crucial role. They perform a variety of functions as the hub of the local network, including as gathering information from cluster member nodes, combining it, and sending the processed data to the base station. In order to maintain efficient information flow, certain cluster head nodes further serve as data relays, transmitting data between cluster heads. These complex operations result in significantly higher energy consumption for cluster head nodes compared to other nodes. As a result, our design specifies that cluster head nodes are simply in charge of data aggregation and do not carry out data slicing tasks.

(1)Determining the Number of Slices

A dynamic slicing technique based on trust values is presented in this study. The core idea of this strategy is to flexibly adjust the number of slices based on trust values, aiming to optimize network performance and data transmission efficiency. Here, the trust value refers specifically to the comprehensive trust assessment made by the cluster head node towards its cluster members. This assessment is multidimensional, considering various critical factors such as link quality, historical behavior of nodes, and their current energy status.

In the dynamic slicing strategy, the level of trust directly determines the number of slices during node data transmission. Specifically, for nodes with higher trust values, we tend to believe that they can complete data transmission tasks more stably and reliably. Therefore, to reduce network complexity and unnecessary overhead, we decrease the number of slices for these nodes. Conversely, for nodes with lower trust values, due to their potential uncertainties and risks, we increase the number of slices to enhance the fault tolerance of data transmission. This way, even if a slice encounters issues during transmission, other slices can still carry partial or complete information to the destination.

Overall, this trust-based dynamic slicing strategy aims to ensure data transmission security, reduce network traffic, and improve overall data transmission efficiency through intelligent adjustment of slices. The specific slicing rules can be represented by the following formula:(5)$$N_{s}^{i}=g(T_{i},N_{n}^{i})=\left\{\begin{array}{ll} \begin{array}{l} \min(2,N_{n}^{i})\\ \min(3,N_{n}^{i})\\ \min(4,N_{n}^{i}) \end{array} & \begin{array}{l} 0.8\leq T_{i}<1\\ 0.6\leq T_{i}<0.8\\ 0.4\leq T_{i}<0.6 \end{array} \end{array}\right.$$
where $T_{i}$ is the cluster head node’s trust value for its cluster member node i, $N_{s}^{i}$ is the number of slices for node i, and $N_{n}^{i}$ is the number of node i’s adjacent nodes. Different numbers of slices are set for different trust values. Since low-trust nodes are not selected as cluster members during the dynamic clustering phase, the trust values of cluster members are generally within a reasonable range. Furthermore, a node’s number of slices cannot be greater than the total number of nodes that surround it. This fine-grained slicing strategy ensures that nodes with different trust values transmit data optimally.

(2)Determining the Size of Slices

In determining the size of data slices, this paper proposes an innovative strategy that bases the slice size on the distance of data transmission. The objective of this approach is to enable cluster members to intelligently divide the data to be sent into slices of varying sizes, depending on the distance to their neighboring nodes. By using this approach, we may improve the efficiency and dependability of data transmission by better accommodating variations in the distance between network nodes.

More specifically, a node determines how far away its neighbors’ nodes are before deciding to transfer data. For nodes that are farther away, the sending node opts to transmit smaller data slices. The advantage of this is that smaller slices are more likely to maintain data integrity during transmission. Additionally, due to their smaller size, the cost of retransmission is relatively low. Moreover, smaller data slices allow for more efficient utilization of network resources, reducing energy consumption associated with long-distance transmission. Conversely, for nodes that are closer, the sending node chooses to send larger data slices. Because of the proximity, both the reliability and efficiency of transmission are relatively high, enabling larger slices to convey more information more effectively. This strategy not only reduces the number of data packets, lowering the likelihood of network congestion, but also increases overall data throughput. Overall, this strategy of determining slice size based on data transmission distance effectively adapts to neighboring nodes at varying distances, optimizing data transmission within the network.

We define $M_{i}$ as the initial data that node i needs to transmit. When node i sends data to node j, the size of the data slice sent is denoted as $m_{ij}$. Specifically, when node i retains a data slice for itself, we refer to it as $m_{ii}$. Meanwhile, we use $d_{ij}$ to represent the physical distance between node i and node j. Based on a dynamic slicing strategy that considers trust values, node i will divide its data to be sent into $N_{s}^{i}$ slices, one of which is $m_{ii}$. The size of $m_{ii}$ will be determined according to the specific strategy.
(6)$$m_{ii}=\frac{1}{N_{s}^{i}}\times M_{i}$$

This represents the separation between node i and node j, which is nearby:(7)$$d_{ij}=\sqrt{{\left(x_{i}-x_{j}\right)}^{2}+{\left(y_{i}-y_{j}\right)}^{2}+{\left(z_{i}-z_{j}\right)}^{2}}$$

Then, the sizes of the remaining $\left(N_{s}^{i}-1\right)$ slices can be represented as:(8)$$m_{i1}:m_{i2}:\cdots :m_{ij}:\cdots m_{i(N_{s}^{i}-1)}=\frac{1}{d_{i1}}:\frac{1}{d_{i2}}:\cdots :\frac{1}{d_{ij}}:\cdots :\frac{1}{d_{i(N_{s}^{i}-1)}}$$
where $d_{i1}<d_{i2}<\cdots <d_{ij}<\cdots <d_{i(N_{s}^{i}-1)}$ and the distance between node i and any nearby node j can be expressed as follows if the shortest distance is used as the unit distance:(9)$$d_{ij}=u_{j}\times d_{i1}$$

Then, Equation (9) can be transformed into:(10)$$m_{i1}:m_{i2}:\cdots :m_{ij}:\cdots m_{i(N_{s}^{i}-1)}=1:\frac{1}{u_{2}}:\cdots :\frac{1}{u_{i}}:\cdots :\frac{1}{u_{\left(N_{s}^{i}-1\right)}}$$

Consequently, the network’s cluster member nodes can use the following equation to calculate the size of their slice data:(11)$$m_{ij}=\frac{u_{N_{s}^{i}-j}}{1+\sum_{j=2}^{N_{s}^{i}-1}u_{j}}\times \left(\frac{N_{s}^{i}-1}{N_{s}^{i}}\right)\times M_{i}$$

This strategy makes sure that each slice uses the least amount of energy possible when transmitting data by sizing the slices according to the transmission distance. Dynamic slicing technology also enhances network security. Since attackers cannot obtain all information from only a small portion of original data, even if they intercept some data slices, they cannot easily reconstruct the complete original data. The dynamic slicing process is shown in Figure 4.

### 4.3. Data Mixing and Aggregation

During the data mixing phase, cluster member nodes combine the data slices they have retained with the data slices they have received after dividing the data slices according to the dynamic slicing algorithm and sending them. This creates a new mixed data packet. During this process, the exchange of data slices between nodes may cause congestion in communication links, leading to the risk of data collision and information loss. Therefore, it is necessary to formulate a reasonable slicing transmission strategy. In the proposed scheme, except for retaining one data slice themselves, nodes transmit the remaining slices to neighboring nodes based on their trust values, prioritizing high-trust neighboring nodes as recipients to ensure data security. Additionally, for nodes requiring four slices, one slice needs to be sent to the nearest neighboring cluster node. This strategy helps reduce data slice collisions and improves the accuracy of data aggregation, as shown in Figure 5.

During data aggregation, each cluster member node encrypts mixed data packets using a shared key before transmitting them to the cluster head node. The cluster head node receives the encrypted mixed data packets and decodes them with the shared key. It then constructs the final data packet for the cluster by merging the decrypted data with the data it retains. The final data packet within the cluster is sent to the next cluster head node until it reaches the base station, at which point the cluster head node travels toward the base station. Figure 6 illustrates a process illustration.

## 5. Simulation Experiment and Analysis

In order to assess the suggested TMSDA algorithm’s performance in-depth in this work, we conducted a detailed comparative research with two popular algorithms, SMART [6] and EEHA [7]. This experiment focused on four core performance indicators: communication load, energy consumption, aggregation accuracy, and data security. As a traditional data aggregation technique built on slicing technology, the SMART algorithm serves as a crucial baseline for our investigation. The EEHA algorithm, on the other hand, further optimizes and improves the data slicing strategy for cluster head nodes based on SMART.

The overall comparison of TMSDA with related works [6,7,30,31] is shown in Table 1. From Table 1 (√ indicates possessing the property, × indicates not possessing the property), it can be seen that the proposed protocol can guarantee more security features than the related works.

In addition, in this experiment, we employed the adaptive trust evaluation model from reference [32], which can provide reliable node trust values in complex underwater environments. This model first evaluates the link quality between nodes based on communication conditions. Then, it dynamically adjusts the weights of three direct trust factors using a variable weight algorithm to quickly reflect changes in direct trust. Indirect trust is obtained by adjusting the recommendation trust weights of neighboring nodes based on deviation. The trust values of the historical and present cycles are then combined to determine the node’s final trust value. To simulate a real network environment, we used Matlab (version R2020b) to build an experimental simulation environment.

### 5.1. Communication Load

Due to the complex environment of underwater nodes, communication load is a crucial performance indicator in underwater wireless sensor networks. This experiment focuses on the communication load primarily generated during the dynamic slicing phase, where member nodes within a cluster perform data slicing and exchange with each other. This process significantly increases the communication load. In the SMART algorithm, every node in the network performs data slicing and exchange operations with a fixed number of slices. Although this approach is simple, it may result in considerable communication overhead. The EEHA algorithm resolves the SMART method’s problems with excessive data slicing and the cluster head nodes’ high energy consumption. In EEHA, intermediate nodes no longer perform slicing operations but only data aggregation and forwarding, while only leaf nodes perform slicing. This enhancement drastically cuts down on communication overhead and the quantity of data slices. Adding to this, the slicing method based on EEHA is further optimized by the TMSDA algorithm suggested in this research. Based on each node’s trust value and transmission distance, the algorithm can dynamically modify the number of slices and slice size. These upgrades can further lower the overhead associated with network connectivity.

The SMART algorithm assumes that every node has three slices by default. The EEHA algorithm assumes that leaf nodes make up 80% of all nodes in the network and that each leaf node has three slices by default. In the TMSDA algorithm, slicing is performed based on the trust-based dynamic slicing strategy proposed in this paper. To validate the effectiveness of our algorithm, this experiment provides a detailed comparison of the number of slices between SMART, EEHA, and TMSDA algorithms at different time intervals.

It is clear from Figure 7’s comparison of the three algorithms’ data slice counts that the TMSDA algorithm is capable of significantly lowering the network’s communication load. The cluster head nodes in the TMSDA algorithm do not take part in data slicing operations; only the member nodes within the cluster do. Compared to the SMART algorithm, where all nodes must execute data slicing, this architecture yields a far smaller amount of data slices in the network. Additionally, based on the trust value that the cluster head assigns to each member node, the TMSDA algorithm provides an adaptive trust evaluation model that dynamically decides the number of slices for each node. This mechanism effectively filters out nodes with lower trust values, thereby reducing the proportion of nodes participating in slicing and further minimizing communication overhead. In contrast, the EEHA algorithm assumes that all intermediate nodes participate in slicing by default, without considering their trustworthiness. The TMSDA algorithm, with its focus on trust-based slicing, demonstrates superior efficiency in managing communication load within the network.

### 5.2. Energy Consumption

The design of data aggregation algorithms must consider the energy consumption of nodes, as they have limited energy resources. Nodes in underwater wireless sensor networks primarily need energy for data transmission and internal computation. It is important to note that node computation uses comparatively less energy than data transfer. To simplify the experiment and facilitate comparative analysis, this study focuses on the energy consumption during the process of sending and receiving data by nodes. To comprehensively verify the effectiveness of the TMSDA algorithm, we conducted exhaustive comparative experiments on energy consumption, involving three algorithms: SMART, EEHA, and TMSDA.

Figure 8 illustrates that the SMART algorithm uses more energy than the EEHA and TMSDA algorithms. This is because only leaf nodes—not aggregating nodes—participate in data slicing and exchange in the SMART algorithm, while all nodes—not aggregating nodes—do so in the EEHA and TMSDA algorithms. Consequently, as compared to the SMART algorithm, the latter two systems demonstrate reduced energy consumption. Moreover, the TMSDA algorithm uses the least amount of energy. This is primarily due to the fact that the TMSDA algorithm has a smaller number of slices, resulting in less data exchange between nodes. Additionally, the TMSDA algorithm determines the slice size based on the data transmission distance, allowing for more efficient data transmission and, consequently, lower energy consumption. This extends the network’s lifetime. Therefore, the TMSDA algorithm demonstrates superior energy efficiency compared to both the SMART and EEHA algorithms.

### 5.3. Aggregation Accuracy

The aggregation accuracy is mainly measured by comparing the actual aggregated data with the theoretical data. However, in practical applications, issues such as collisions, delays, and retransmissions during data transmission can affect the accuracy of data aggregation. The higher the amount of data transmitted per unit time, the greater the probability of encountering these negative impacts, resulting in poorer data aggregation accuracy. The simulation experiment results are shown in Figure 9.

From Figure 9, it can be seen that as the data aggregation time increases, the TMSDA algorithm exhibits higher aggregation accuracy due to its low communication overhead, significantly outperforming the EEHA and SMART algorithms. The aggregation accuracy of the SMART algorithm is relatively low, mainly because its large data communication volume leads to frequent data collisions, which affect its aggregation accuracy. However, as time passes, the probability of data packet collisions and losses decreases, resulting in improved aggregation accuracy for all three algorithms. Additionally, the TMSDA algorithm incorporates a trust mechanism that considers the impact of link quality, effectively reducing the influence of negative factors on aggregation accuracy. Furthermore, the TMSDA algorithm has fewer data slices compared to the other schemes, which relatively reduces data collisions and enhances aggregation accuracy performance.

### 5.4. Data Security

To prevent malicious nodes from eavesdropping on sensitive data from internal neighboring nodes, the TMSDA algorithm employs a data slicing strategy based on a trust mechanism. Under this strategy, the number of slices is not fixed but is dynamically determined based on the trust values that the cluster head assigns to the cluster members. This design makes it difficult for attackers to accurately determine the number of slices, thereby increasing the difficulty of recovering the original data. Additionally, in the TMSDA algorithm, cluster members prioritize sending data slices to neighboring nodes with higher trust values, further enhancing data security. The simulation results, as shown in Figure 10, compare the data security performance of the three algorithms.

Figure 10 compares the data security performance of three algorithms, and the results show that the TMSDA algorithm exhibits a lower probability of private data leakage compared to the SMART and EEHA algorithms. The main reason for this is that the TMSDA algorithm significantly reduces the volume of data communication by reducing the number of nodes’ private data slices, which lowers the possibility of data eavesdropping. Furthermore, the TMSDA method does not set a fixed number or size for each node’s slices. This characteristic makes it difficult for malicious nodes to accurately determine whether they have acquired all the necessary slices, even if they successfully eavesdrop on a node’s data slice. As a result, they cannot effectively recover the original data. Therefore, the TMSDA algorithm effectively enhances network security by reducing data communication volume and increasing slice dynamism.

## 6. Conclusions

Based on a trust mechanism, this research proposes a secure data aggregation approach to address the issues of high communication cost and low aggregation accuracy in existing data aggregation methods. The cluster head nodes are first selected by the algorithm based on the local density of nodes in the network. The cluster head nodes then use the trust ratings of neighboring nodes to choose nodes with high trust values as their cluster members to complete the cluster building process. Afterward, cluster member nodes dynamically determine the number and size of data slices based on the trust value assigned to them by the cluster head node and their distances from other neighboring nodes. This strategy not only significantly enhances data security but also effectively reduces communication consumption.

Although the scheme proposed in this paper has successfully achieved the expected goals, there are still some limitations that require further improvement and refinement. For instance, the secure data aggregation scheme designed in this paper performs well in ensuring data security. However, it has not yet addressed the issue of data integrity verification. In future research, we plan to conduct in-depth studies on problems such as data loss recovery and data error correction to further improve the scheme proposed in this paper.

In addition, in practical applications, wireless sensor nodes are often deployed in harsh, unattended, or even hostile environments, making them vulnerable to capture and compromise. Attackers can use information, such as the compromised node’s key, to impersonate legitimate nodes and launch internal attacks on the network. Therefore, how to quickly identify compromised nodes in the network has become a very important research direction in the field of wireless sensor network security research.

## Figures and Tables

**Figure 1 sensors-24-04352-f001:**
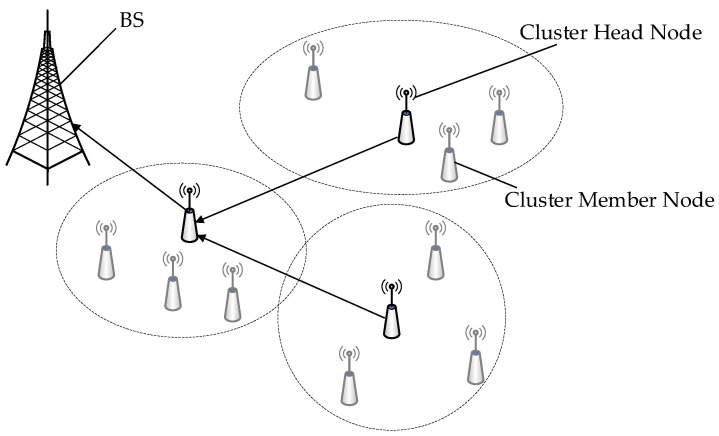
Network structure.

**Figure 2 sensors-24-04352-f002:**
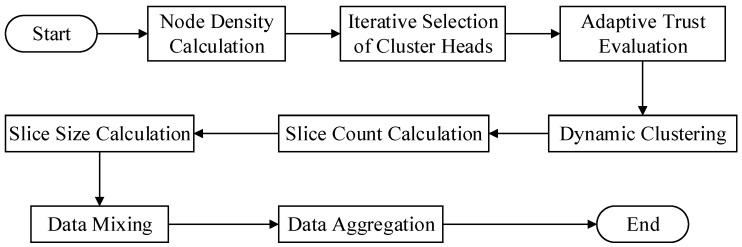
TMSDA algorithm.

**Figure 3 sensors-24-04352-f003:**
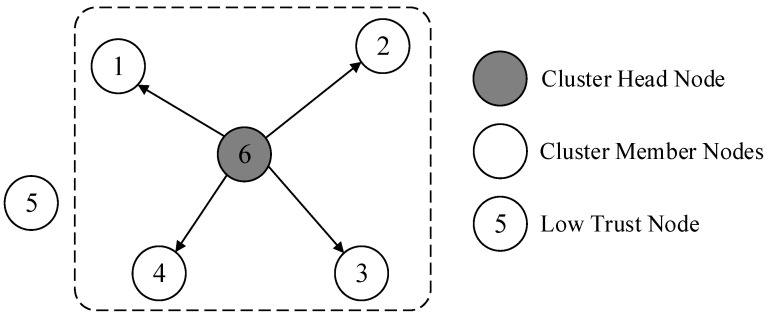
Dynamic clustering.

**Figure 4 sensors-24-04352-f004:**
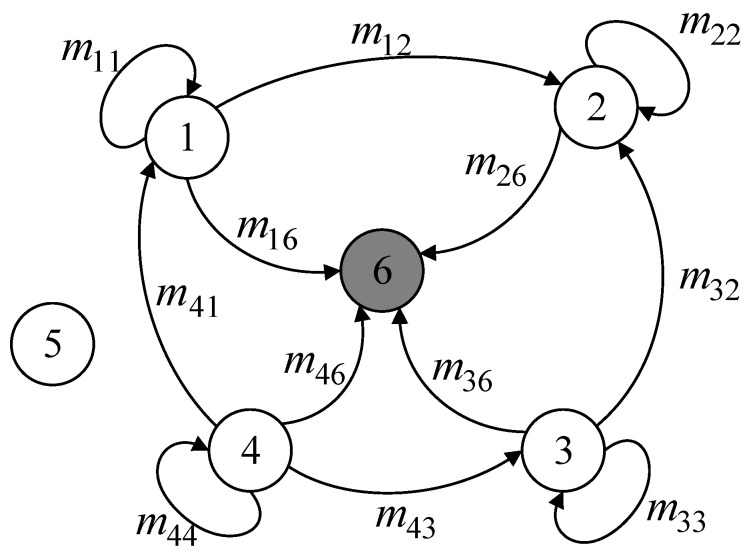
Dynamic slicing.

**Figure 5 sensors-24-04352-f005:**
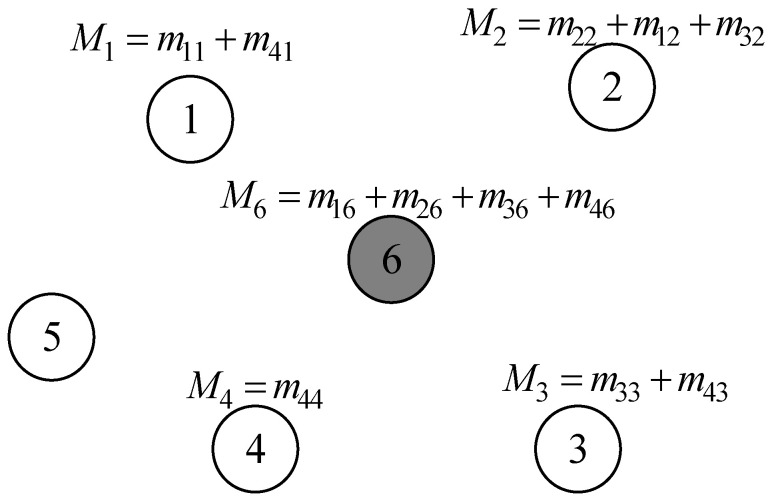
Data mixing.

**Figure 6 sensors-24-04352-f006:**
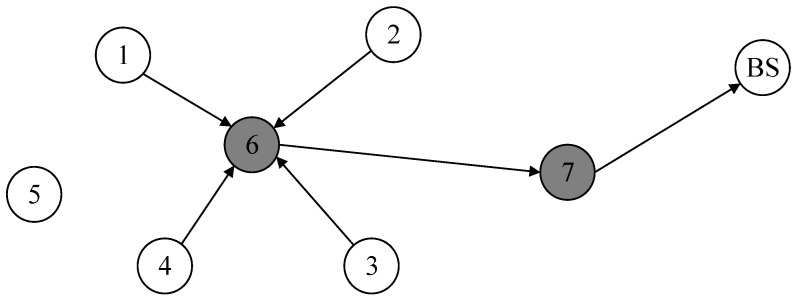
Data aggregation.

**Figure 7 sensors-24-04352-f007:**
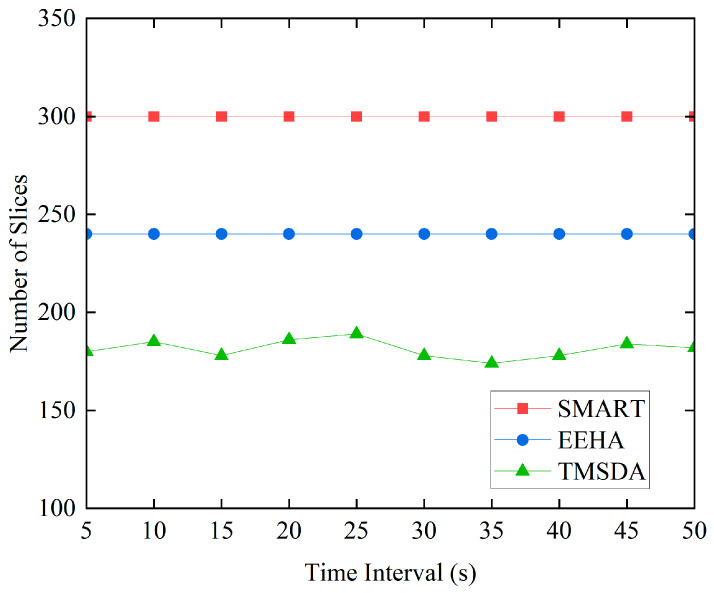
Comparison of the number of slices.

**Figure 8 sensors-24-04352-f008:**
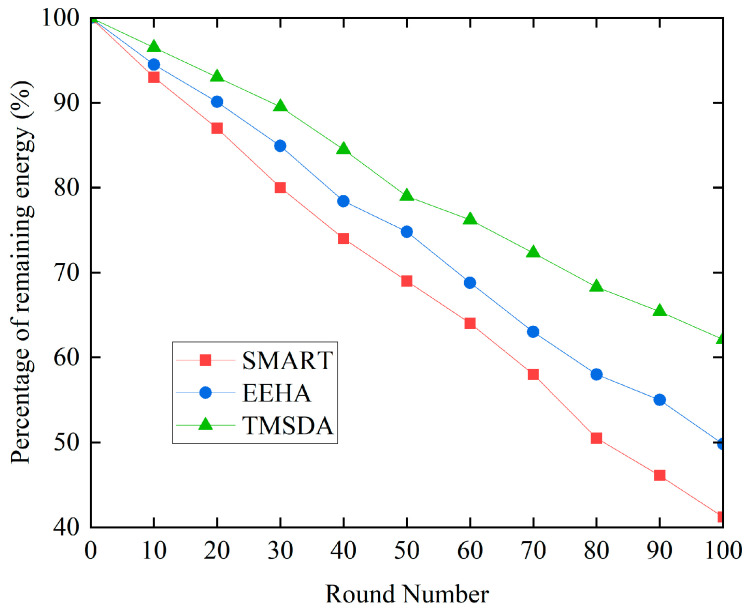
Comparison of energy consumption.

**Figure 9 sensors-24-04352-f009:**
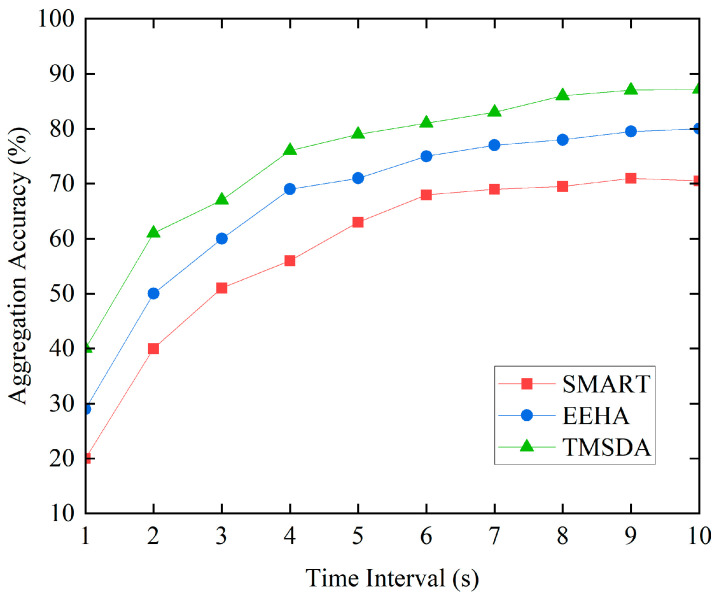
Aggregation accuracy.

**Figure 10 sensors-24-04352-f010:**
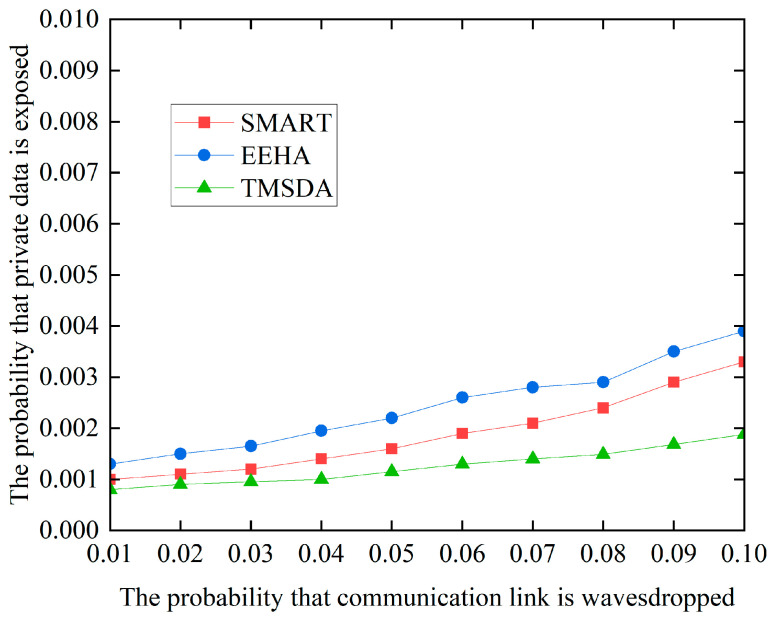
Data security performance.

**Table 1 sensors-24-04352-t001:** Overall comparison with related works.

	[6]	[7]	[30]	[31]	TMSDA
Efficient communication	√	√	√	√	√
Confidentiality	√	√	√	√	√
Reliability	×	×	√	√	√
Dynamic slicing	×	×	×	×	√

## Data Availability

Data are contained within the article.

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
