# Peer review of "A Secure Data Aggregation Algorithm Based on a Trust Mechanism"

_sensors, 2024, doi:10.3390/s24134352_

Round 1
Reviewer 1 Report
Comments and Suggestions for Authors
This paper proposes a secure data aggregation algorithm based on trust for underwater communication.
- I noticed all introductions don't have a single reference. Authors should support their argument by adding references such as Moreover, given the underwater 32
environment's dynamic nature, current solutions often lack the flexibility to handle situations and Although slicing technology can effectively 40
defend against various malicious attacks,
-some references have some mistakes such as [3], [4], [6].
- The literature review relies on an outdated reference. That review needs to be updated to properly reflect a real research gap.
- What is the difference between the contribution in this paper and Ref[22]? It is crucial to identfy which parts are novel compared to the state-of-the-art
- the research gap is not clear and the literature review is not properly written based on different themes or types.
- I strongly suggest adding a figure for the network model Section3 to highlight the workflow of the network model
- Why are Refs [21] and [22] in the literature review section?
- Fig 3 is not cited within the context.
- Is the dynamic slicing based on the Markov model?
- EEHA and SMART are not expanded in the context
- How is the trust rating aggregated?
Comments on the Quality of English Language
Overall the writing is good but there are some minor typos such as the word mixes is not reflect the content
Author Response
Thanks for your suggestions, the response is attached.

Reviewer 2 Report
Comments and Suggestions for Authors
This article proposes a Secure Data Aggregation Algorithm Based on Trust Mechanism and discusses the unique characteristics of underwater environments. The article dynamically evaluates the trust of nodes and adjusts the number and size of node slices based on trust values and transmission distances, thereby reducing network communication costs and improving the accuracy of data aggregation. In addition, the article also introduces the construction methods of network models and secure transmission channels, as well as the mechanism for establishing secure connections based on random key distribution. These contents have certain research value for solving security and data aggregation problems in underwater wireless sensor networks. However, I believe that there are still some issues that need to be addressed in this article:
1. The article mentions a network model based on tree and cluster topology, but the selection and advantages of this structure have not been fully explained and demonstrated. The author can further explore why this hybrid topology is chosen and analyze its impact on data transmission.
2. This article proposes a random key distribution mechanism, but does not provide a detailed explanation of its advantages and applicability, and compares it with other key distribution methods to further demonstrate its effectiveness.
3. The article did not discuss actual application scenarios and potential application prospects. In the conclusion section, the practical application of the algorithm should be discussed, including possible application areas, challenges, and future research directions.
4. The last section should specifically discuss the limitations and potential future extensions of this algorithm.
Author Response

(The authors gave the same response as above.)

Reviewer 3 Report
Comments and Suggestions for Authors
1. There are too few references, please add references in the introduction section.
2. In the Network Model, add an example diagram to clearly show the network topology.
3. The node degree is the number of nodes in the communication range, and does not need to be expressed by a formula.
4. Do you iterate the cluster header based on the number of rounds? If this node runs out of energy and dies, will it iterate? Please give a more direct explanation.
5. Line 417 "The final packet within the cluster will be sent to the next cluster head node". Add the process of sending packets to the next cluster head node in Figure 5.
6. Indicate which references were cited in the two protocols compared.
7. The simulation results do not support your conclusions, and the proposed scenario should be compared with at least two recent scenarios.
Comments on the Quality of English LanguageEnglish requires only a small amount of editing.
Author Response

(The authors gave the same response as above.)

Round 2
Reviewer 2 Report
Comments and Suggestions for Authors
The authors have made comprehensive revisions based on the review comments and are suitable for publication.
Author Response
Comment 1: The authors have made comprehensive revisions based on the review comments and are suitable for publication.
Response: Thank you for the positive feedback. We are glad to hear that our comprehensive revisions based on the review comments have met the criteria for publication.
Reviewer 3 Report
Comments and Suggestions for Authors
The proposed protocol should be compared with at least one recent protocol.
Comments on the Quality of English LanguageI have no further comments.
Author Response
Thanks for your advice. The response is attached.
